# Photogating-assisted tunneling boosts the responsivity and speed of heterogeneous WSe$_2$/Ta$_2$NiSe$_5$ photodetectors

Mingxiu Liu[1,2,6], Jingxuan Wei [3,4,6], Liujian Qi [1,2,6], Junru An[1,2], Xingsi Liu[3], Yahui Li[1,2], Zhiming Shi [1,2], Dabing Li [1,2] ✉, Kostya S. Novoselov [5], Cheng-Wei Qiu [3] ✉ & Shaojuan Li [1,2] ✉

Photogating effect is the dominant mechanism of most high-responsivity two-dimensional (2D) material photodetectors. However, the ultrahigh responsivities in those devices are intrinsically at the cost of very slow response speed. In this work, we report a WSe$_2$/Ta$_2$NiSe$_5$ heterostructure detector whose photodetection gain and response speed can be enhanced simultaneously, overcoming the trade-off between responsivity and speed. We reveal that photogating-assisted tunneling synergistically allows photocarrier multiplication and carrier acceleration through tunneling under an electrical field. The photogating effect in our device features low-power consumption (in the order of nW) and shows a dependence on the polarization states of incident light, which can be further tuned by source-drain voltages, allowing for wavelength discrimination with just a two-electrode planar structure. Our findings offer more opportunities for the long-sought next-generation photodetectors with high responsivity, fast speed, polarization detection, and multi-color sensing, simultaneously.

Photodetectors constitute the base stone of various optical and optoelectronic devices, whose recent development is mainly driven by emerging technologies such as photonic integrated circuits[1–6], the Internet of Things[7–9], and automation[10–13]. Those new technologies keep pushing the photodetectors to acquire higher responsivity, faster response, lower power consumption, and more functionalities such as wavelength and polarization sensitivity. However, the above requirements cannot be simultaneously met in the conventional photodetectors based on bulky materials, such as silicon[14], germanium[15], and III-V semiconductors[16]. A promising solution could be using two-dimensional (2D) materials with strong light-matter interactions, tunable band gaps, compatibility with existing semiconductor production lines, and the rich opportunities provided by atomically sharp heterointerfaces[11,17–19]. For example, 2D materials photodetectors with ultra-high responsivities ($10^3$–$10^7$ A/W) have been well documented, in which the photogating effect dominates the photoresponse[20–23]. The excellent responsivity performance results from the significant photodetection gain: $G = \tau / t_L$, where $\tau$ and $t_L$ are the photocarrier lifetime and carrier transit time, respectively[24]. Unfortunately, such a mechanism is always achieved with a long photocarrier lifetime, thereby suffering prolonged response time[20,22,25,26], with typical values in the millisecond range, unjustified for many applications such as imaging. The trade-off between high responsivity and fast speed is illustrated in Fig. 1a. A closer scrutiny of the photodetection gain

[1]State Key Laboratory of Luminescence and Applications, Changchun Institute of Optics, Fine Mechanics and Physics, Chinese Academy of Sciences, Jilin 130033 Changchun, PR China. [2]University of Chinese Academy of Sciences (UCAS), 100049 Beijing, PR China. [3]Department of Electrical and Computer Engineering, National University of Singapore, Singapore 117583, Singapore. [4]School of Optoelectronic Science and Engineering, University of Electronic Science and Technology of China, 611731 Chengdu, PR China. [5]Institute for Functional Intelligent Materials, National University of Singapore, Singapore 117544, Singapore. [6]These authors contributed equally: Mingxiu Liu, Jingxuan Wei, Liujian Qi. ✉e-mail: lidb@ciomp.ac.cn; chengwei.qiu@nus.edu.sg; lishaojuan@ciomp.ac.cn

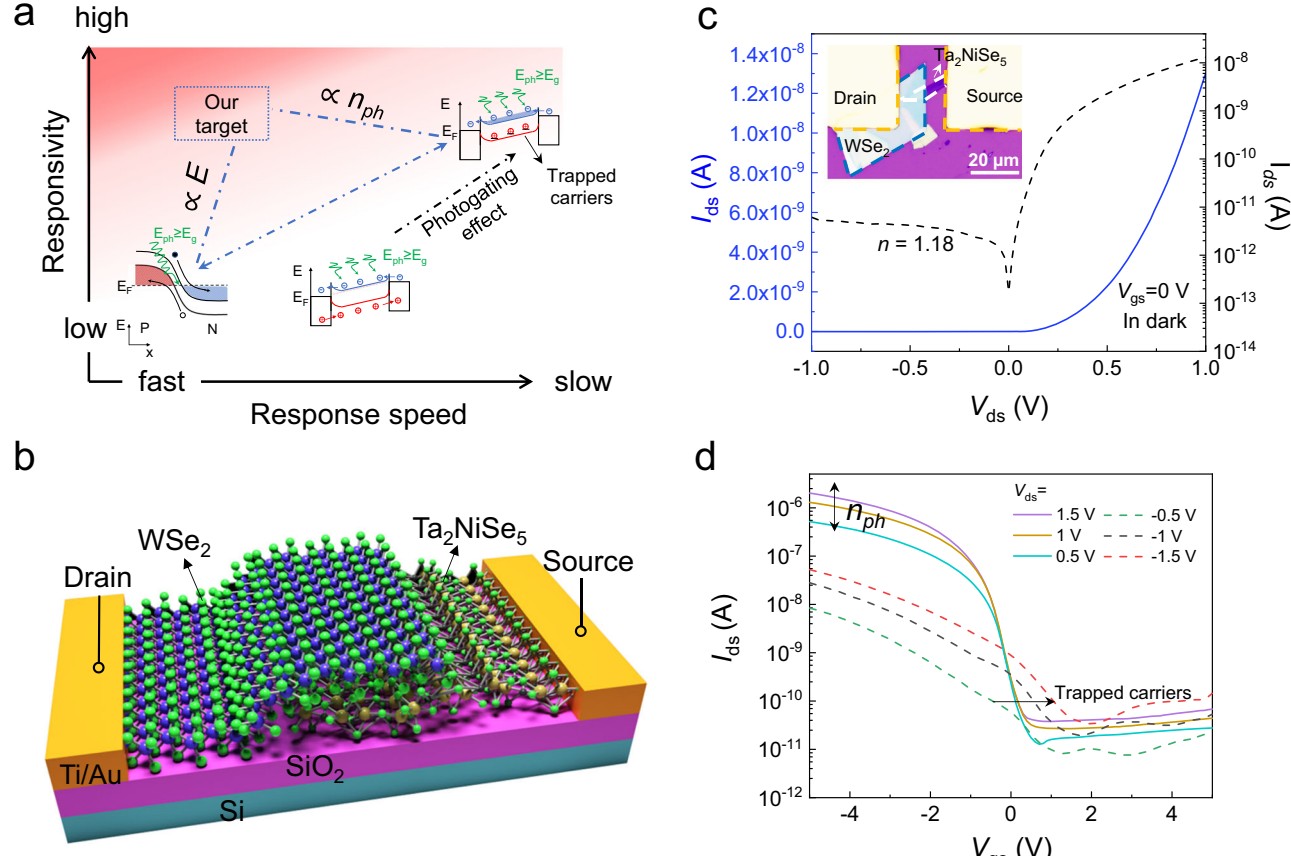

**Fig. 1 | Bias-tunable transport behavior in our 2D heterostructure photo-detector. a** Illustration of the trade-off between the responsivity and speed in 2D materials photodetectors. To overcome the trade-off and enter the upper left area, we conventionally need to increase the power consumption, denoted by the red shade. The energy band diagram on the lower left side is the photovoltaic mode of photodetector, the middle is the photoconductive mode, and the upper right side belongs to photogating mode. Where $E_{ph}$ is the photon energy, $n_{ph}$ is the photo-generated carrier concentration, $E_g$ is the bandgap, $E_F$ is the Fermi level of the material, and $E$ is the external electric field. Different background colors represent the variation trend of possible required power consumption of 2D photodetectors, which increases gradually from the white color at the lower right corner to the red color at the upper left corner of figure. **b** Schematic of a $WSe_2/Ta_2NiSe_5$

heterostructure device. **c** Source-drain I-V curves of a representative device in dark. No gate voltage was applied ($V_{gs} = 0$ V). This device consists of an approximately 60-nm-thick $WSe_2$ flake on top of a 13-nm-thick $Ta_2NiSe_5$ flake. The dashed line represents the logarithmic form of the I-V curve. The ideal factor $n$ is defined as $n = (q/KT) \cdot (dV/d\ln I)$, the inset is an optical microscope diagram of the device. **d** Transfer curves ($I_{ds}$-$V_{gs}$ (back gate voltage)) of the device at different source-drain biases, $V_{ds}$, under the illumination of a white light source. The distinct behaviors between positive (solid lines) and negative (dashed lines) $V_{ds}$ allow us to analyze the underlying photoresponse mechanism. $n_{ph}$ represents the carrier concentration under the light illumination, the horizontal arrow represents the shift of transfer curve induced by the trapped carriers.

formula reveals that the responsivity can also be increased by a shorter transit time. Conventionally, this is achieved using devices with shorter channel lengths or larger bias voltages, which face limitations in practical applications. The channel length cannot be scaled down arbitrarily due to the fabrication capability, increased dark currents and hence power consumption[27–29]; the bias voltage usually keeps at the $V_{dd}$ in the complementary metal–oxide–semiconductor (CMOS) circuitry of around 1 V[30,31]. Furthermore, while polarization and wavelength of light play essential roles in widespread applications, most of the existing photodetectors are only sensitive to the intensity of light[9,32,33]. Therefore, how to overcome the responsivity-speed trade-off and extend the functionalities of photodetectors remains an prominent problem.

In this work, we report a 2D $WSe_2/Ta_2NiSe_5$ heterostructure device which can fulfill all the above requirements. Our analysis suggests that a tunneling process can significantly reduce the transit time in our device, dramatically enhancing the photodetection gain and hence the responsivity. As a result, we have a sufficient responsivity budget to reduce the response time below the threshold of around 1 ms for practical imaging applications by managing the trap sites. Namely, due

to the tunneling enabled short transit time, our device allows high responsivity even at a relatively short carrier lifetime. The responsivity reaches above $10^3$ A/W, and the response time is down to ~1 μs. While the above performance is already superior to the existing 2D photogating devices in the case of imaging applications, our device also shows lower power consumption in the order of nW, broad working range from visible to infrared, and polarization dependence due to the intrinsic anisotropy of $Ta_2NiSe_5$. Interestingly, the anisotropic photoresponse ratio is bias-tunable and wavelength-dependent, providing a promising platform for wavelength discrimination. Our results may pave the way towards the future development of high-performance 2D photodetectors potentially for applications in miniaturized spectroscopy, spectral imaging, objects and threats identification.

## Results

### Bias-tunable transport behavior of $WSe_2/Ta_2NiSe_5$ device

Figure 1b shows the schematic of the $WSe_2/Ta_2NiSe_5$ heterostructure photodetectors. The layered $WSe_2$ and $Ta_2NiSe_5$ flakes were mechanically exfoliated from their bulk materials, and the electrodes were composed of titanium/gold (10/80 nm). As a member of the

ternary chalcogenides, $Ta_2NiSe_5$ has layered monoclinic structures below the transition temperature (328 K)[34–37], where layers formed by periodically assembled $[TaSe_6]_2$ dimer chains and $NiSe_4$ single chains are weakly bonded via van der Waals interactions (Supplementary Fig. 1a). The distorted chain structure leads to a strong in-plane anisotropy and all three lattice constants ($a = 3.5$ Å, $b = 12.8$ Å, and $c = 15.6$ Å) are different[34]. The crystallographic direction can be determined by optical microscopy and angle-resolved polarized Raman spectroscopy in our experiments, in which the long-axis of the exfoliated flake corresponds to the $a$ direction (Supplementary Note 1)[38]. In addition, $Ta_2NiSe_5$ maintains its direct bandgap from 0.36 eV in the bulk to monolayer[39], with a reported high carrier mobility[40]. The atomic structure diagram of $2H$-$WSe_2$ possesses an in-plane isotropic hexagonal symmetry ($a = b = 3.3$ Å) with two layers per repeated unit (Supplementary Fig. 1b). Few-layer $WSe_2$ has a sizeable bandgap of ~1.2 eV, high absorption coefficient and good ambient stability[41,42]. The efficient charge transfer at the $WSe_2/Ta_2NiSe_5$ interface and the high quality of materials were confirmed by the optical measurements (Supplementary Note 2)[43].

Multiple heterostructure devices comprising different thicknesses were investigated (Supplementary Note 3). Figure 1c plots the source-drain current ($I_{ds}$) of a representative device in dark at room temperature (see the inset of Fig. 1c for a top-view optical image). The devices exhibit typical rectification characteristics with an ideality factor ($n$) of ~1. We then measured the optoelectrical properties of the device under the illumination of a white light source. The transfer characteristics ($I_{ds}$-$V_{gs}$) of the device are shown in Fig. 1d. The corresponding output curves ($I_{ds}$ -$V_{ds}$) are depicted in Supplementary Fig. 6. From the curve signatures, we found below phenomena: First, the $WSe_2/Ta_2NiSe_5$ heterostructure exhibits obvious $p$-type transport characteristics and a high on/off ratio of ~$10^4$, which is ascribed to the strong gate-tunability of $p$-type $WSe_2$. The electrical polarity of $Ta_2NiSe_5$ and $WSe_2$ were investigated separately (Supplementary Note 4). The results confirm the weakly $n$-type transport property of $Ta_2NiSe_5$ and prominent $p$-type conductivity of $WSe_2$, which is consistent with previous reports[26,44]. The carrier mobility was estimated as 12.43 cm$^2$/V·s and 4.64 cm$^2$/V·s for $Ta_2NiSe_5$ and $WSe_2$, respectively. Second, the characteristics indicate Schottky barrier-dominated transport at positive bias ($V_{ds}$) condition[26,45]. High output current under positive $V_{ds}$ voltage at the metal source contact confirms the formation of a $p$-type contact that is favorable for hole injection at the $WSe_2$/metal contact, which will be further discussed in detail in the following sections. Third, the device exhibits distinct behaviors under different source-drain bias voltages (Fig. 1d). Under negative $V_{ds}$, photogating can be seen as a horizontal shift in the $I_{ds}$−$V_{gs}$ traces under illumination with increasing the magnitude of $V_{ds}$. This phenomenon suggests that trap states where carriers can reside for long times exist at defects or at the interface of the heterostructure[46,47]. However, under positive $V_{ds}$, horizontal shift in the $I_{ds}$−$V_{gs}$ traces was largely suppressed, and increment of the number of photocarriers ($n_{ph}$) under higher positive $V_{ds}$ leads to larger output current. The above observations manifest the bias-tunable transport behavior of the $WSe_2/Ta_2NiSe_5$ heterostructure.

## Photogating-assisted tunneling in $WSe_2/Ta_2NiSe_5$ device

To further illustrate the bias-tunable behavior of the device, we plot the photocurrent mapping and corresponding schematic band diagrams of a typical device in Fig. 2. The device consists of two metal/semiconductor junctions at source/drain side and one 60-nm-thick $WSe_2$/13-nm-thick $Ta_2NiSe_5$ heterojunction. The spatially resolved photocurrent mapping with the illumination of 532 nm laser ($V_{gs} = 0$ V) at various $V_{ds}$ reveals significantly different photocurrent generation images (Fig. 2a, b). Similar results were acquired under 633 nm laser illumination (Supplementary Note 5). The pronounced photocurrent generation in the overlapped $WSe_2/Ta_2NiSe_5$ region is observed at

$V_{ds} = -1$ V. In contrast, when the device is forward biased, the maximum photocurrent is generated near the junction region where the metal electrode contacts $WSe_2$ (Fig. 2b). These observations imply the bias-tunable operation mechanisms in the $WSe_2/Ta_2NiSe_5$ heterostructure, attesting the distinct transport behavior in above Fig. 1d.

The band alignments of few-layer $WSe_2/Ta_2NiSe_5$ heterostructures were investigated by using density functional theory (DFT) methods. The Perdew–Burke–Ernzerhof (PBE) functional was adapted to describe electronic exchange-correlation interaction since its great reliability to transition metal dichalcogenides (TMDCs) including $MoS_2$, $PdSe_2$, $Pd_2Se_3$, $InSe$ and so on[48,49]. The projected band structures of 1–6 Layers $WSe_2/Ta_2NiSe_5$ heterostructures are shown in Supplementary Note 6. The conduction band minimum (CBM) and valence band maximum (VBM) of $Ta_2NiSe_5$ embed in the bandgap of $WSe_2$ despite of the thickness, indicating the type-I alignment. For thicker heterostructures shown in Fig. 2a, b, the relative positions of the conduction band minimum (CBM) and the valence band maximum (VBM) of thick $WSe_2$ (60-nm thick) are ~ 4.0 eV and 5.2 eV, respectively, with a bandgap of 1.2 eV[50,51]. According to recent experimental studies on the energy band of $Ta_2NiSe_5$[40,52], the CBM and VBM of thicker (13-nm thick) $Ta_2NiSe_5$ are ~ 4.6 eV and 4.93 eV with a bandgap of 0.33 eV. The offset of Fermi levels between the $WSe_2$ and $Ta_2NiSe_5$ is measured to be ~111.5 meV (Supplementary Note 5). The band diagram of the device are then determined as illustrated in Supplementary Note 5, which shows the formation of a type-I heterojunction as well.

Accordingly, the energy band diagram when the device is negative biased is depicted in Fig. 2c. Since the external bias direction is identical to the built-in electric field in $WSe_2/Ta_2NiSe_5$ heterojunction, the enhancement of the electric field leads to an efficient separation of photogenerated electron-hole pairs when the incident light irradiates at the heterojunction region. On the contrary, when the device is forward biased. Since the built-in electric field of $WSe_2/Ta_2NiSe_5$ heterojunction is opposite to the external electric field applied across the device, photoexcited-electrons in the $Ta_2NiSe_5$ are drifted towards the interface of $WSe_2/Ta_2NiSe_5$ and accumulated at the heterojunction interface due to the large electron barrier (Fig. 2d). In this case, the trapped electrons at the interface acts as a negative gate, inducing a photogating effect. As a feedback of this behavior, the Fermi level of $WSe_2$ would shift downwards, so the hole concentration increases, fostering the tunneling of hole carriers from drain electrode to $WSe_2$ which are then transported to source electrode. It is noteworthy that the tunneling of carriers have been proved to be feasible in improving the device response speed in recent reports[53]. To verify the carrier tunneling in our device, the photoresponse curves under light illumination of different wavelengths were measured (Fig. 2e, f and Supplementary Fig. 10). The curves at positive bias voltages are well modeled by a tunneling barrier with the Simmons approximation (Fig. 2g). The fitting plot of ln ($I/V^2$) versus $1/V$ shows linear dependence with a negative slope under larger $V_{ds}$, and rises exponentially under small $V_{ds}$ under light illumination, as demonstrated in the figures. Thereby, the dominant tunneling occurs with direct tunneling (DT) at low bias voltage and Fowler–Nordheim tunneling (FNT) at higher voltage (see detailed analysis in Supplementary Note 7)[45,54]. The above observations indicate that the tunneling-dominated transport of charge carriers at the interface of metal/$WSe_2$ dominates the device behavior under positive biases.

To further explore the effect of incident light on the tunneling barrier of the device, we calculated the change of carrier concentration in the material under different incident light intensities. The change in carrier concentration can be estimated by $\Delta n (p) = \frac{C_g \Delta |V_{th}|}{q}$[45,55], where $q$ is electronic charge ($1.6 \times 10^{-19}$ C), $C_g$ is the gate capacitance ($1.23 \times 10^{-8}$ F cm$^{-2}$ for 300 nm $SiO_2$), $V_{th}$ is the threshold voltage, which can be extracted from the transfer curve of the heterojunction device in Fig. 2f and Supplementary Fig. 10. The change in the hole concentration ($N_d$) is expressed as $\Delta N_d = \frac{\Delta n(p)}{t}$, where $t$ represents the material

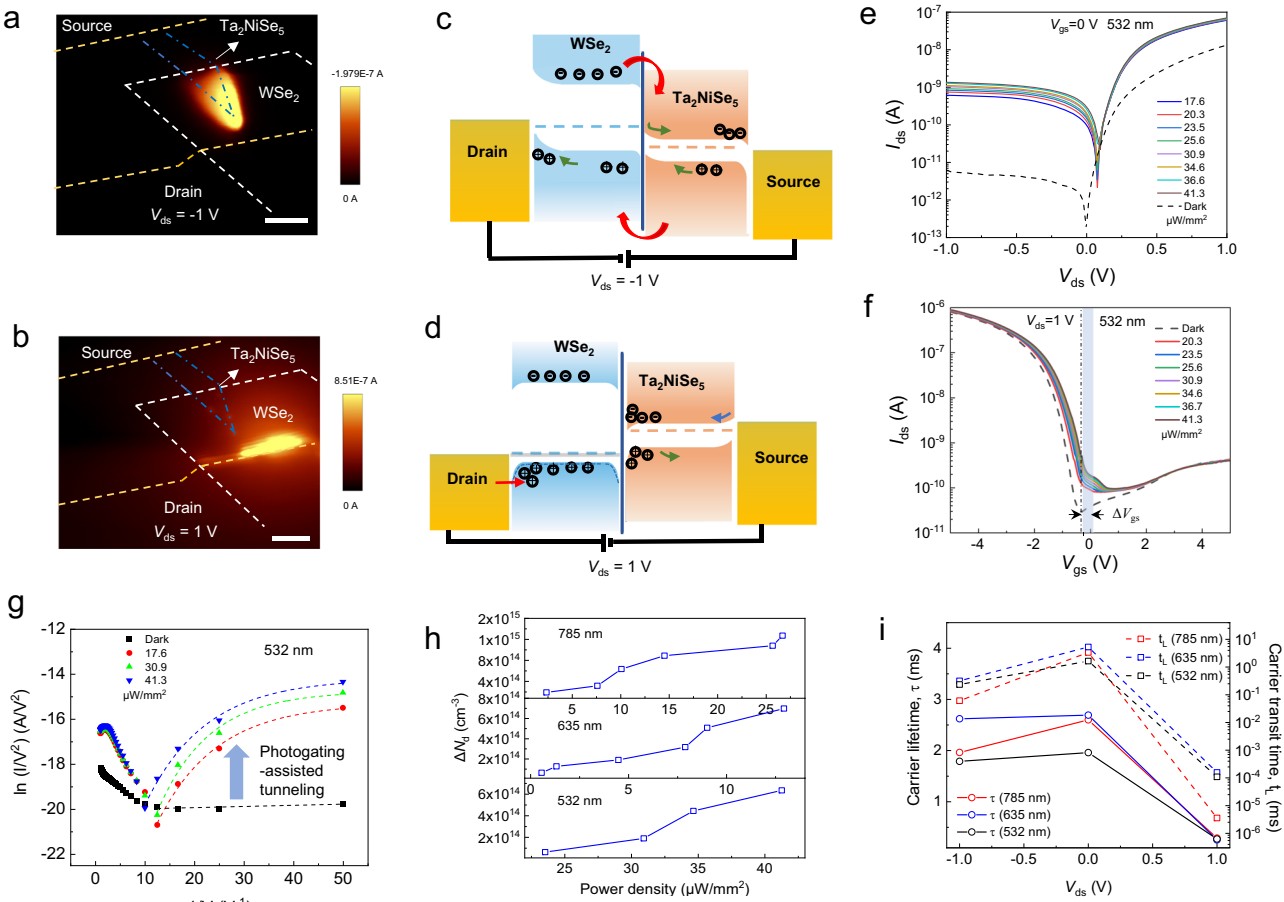

**Fig. 2 | The photogating-assisted tunneling in the WSe₂/Ta₂NiSe₅ hetero-structure.** The spatially resolved photocurrent mapping images at (**a**) $V_{ds}$ = −1 V and (**b**) $V_{ds}$ = 1 V. The scale bar is 7 μm. The illumination wavelength is 532 nm. The white, blue and yellow dashed lines indicate the regions of WSe₂, Ta₂NiSe₅ and metal electrodes, respectively. The energy band diagrams of the WSe₂/Ta₂NiSe₅ heterostructure device at (**c**) $V_{ds}$ = −1 V and (**d**) $V_{ds}$ = 1 V, respectively. The arrow indicates the direction of carrier transport, and the horizontal dashed line represents the Fermi level of the material. **e** Output curves ($I_{ds}$-$V_{ds}$) in dark and under illumination with different power densities. The incident light is at 532 nm

wavelength, and $V_{gs}$ = 0 V. **f** Transfer curves ($I_{ds}$-$V_{gs}$) of the device at $V_{ds}$ = 1 V. $\Delta V_g$ and the shaded area represent the range of the change of the charge neutrality point. **g** Fowler–Nordheim plots of the device at positive $V_{ds}$ in dark and under illumination. The dashed lines are the fits to the experimental data. **h** The changes in hole density ($\Delta N_d$) in the device as the increase of incident light power at different wavelengths. $V_{ds}$ = 1 V. The solid lines are drawn to guide the eye. **i** The extracted photocarrier lifetime (solid lines) and carrier transit time (dashed lines) of the device under different $V_{ds}$. The solid and dashed lines are drawn to guide the eye.

thickness, which is 60 nm for WSe₂ in the device investigated. It is obvious from the Supplementary Fig. 10 that as the incident light intensity increases, $V_{th}$ gradually moves towards positive gate voltage. According to the change of $V_{th}$, we calculated the change of carrier concentration for different incident light power, as shown in Fig. 2h. Based on this, we can conclude that $N_d$ increases as promoting the optical power density. The tunneling barrier width, $d$, is reduced accordingly, following the relationship of $d \propto 1/N_d$[56], thus a thinner barrier width or a higher tunneling probability further assists the tunneling of carriers, leading to a photogating-assisted tunneling in the device.

Based on above, a simultaneous increase in device responsivity and response speed under positive $V_{ds}$ are expected due to the photogating-assisted tunneling effect. On one hand, the responsivity scales with the gain, which is significantly enhanced in our device due to the reduction of transit time via the tunneling process; On the other hand, the carrier lifetime is also reduced by managing the trap sites which assists the recombination of photon-excited carriers, leading to a fast response speed. In other words, a large number of photo-generated electrons are blocked in the Ta₂NiSe₅ conduction band by the barrier of energy band under positive bias, which act as the role of shallow trap sites that capture photogenerated carriers with a shorter

lifetime. In comparison, the intrinsic carrier traps with both shallow and deep energy levels in the materials plays a dominant role under negative bias, featuring a relatively longer lifetime. The measurements of the carrier transit time and photocarrier lifetime of the device under different $V_{ds}$ is shown in Fig. 2i, in good accordance with above analysis (Details on the extraction process are illustrated in Supplementary Note 8).

**Responsivity-speed relations of WSe₂/Ta₂NiSe₅ device**
To look into the responsivity-speed relations, we further analyze the optoelectrical properties of WSe₂/Ta₂NiSe₅ heterostructure devices under monochromatic light illuminations (see details in Supplementary Note 9). According to the power-dependent photoresponse, the responsivities, $R$, against power density are acquired at different $V_{ds}$ (Fig. 3a). Notably, a high responsivity is obtained with $R = 2.2 \times 10^4$ A/W (780 nm, 0.05 μW/mm²) when $V_{ds}$ = 1 V is applied, which is orders of magnitudes higher than that at $V_{ds}$ = −1 V (8.8 A/W). Similar enhancement at positive $V_{ds}$ is also observed in multiple devices (Fig. 3c). The $R$ values are decreased with promoting light power at $V_{ds}$ = ±1 V, and this phenomenon was generally attributed to the shortened photoinduced carrier lifetimes by Auger processes or by the saturation of trap states under a high photon flux[21,57]. The device noise spectral densities ($S_n$) is

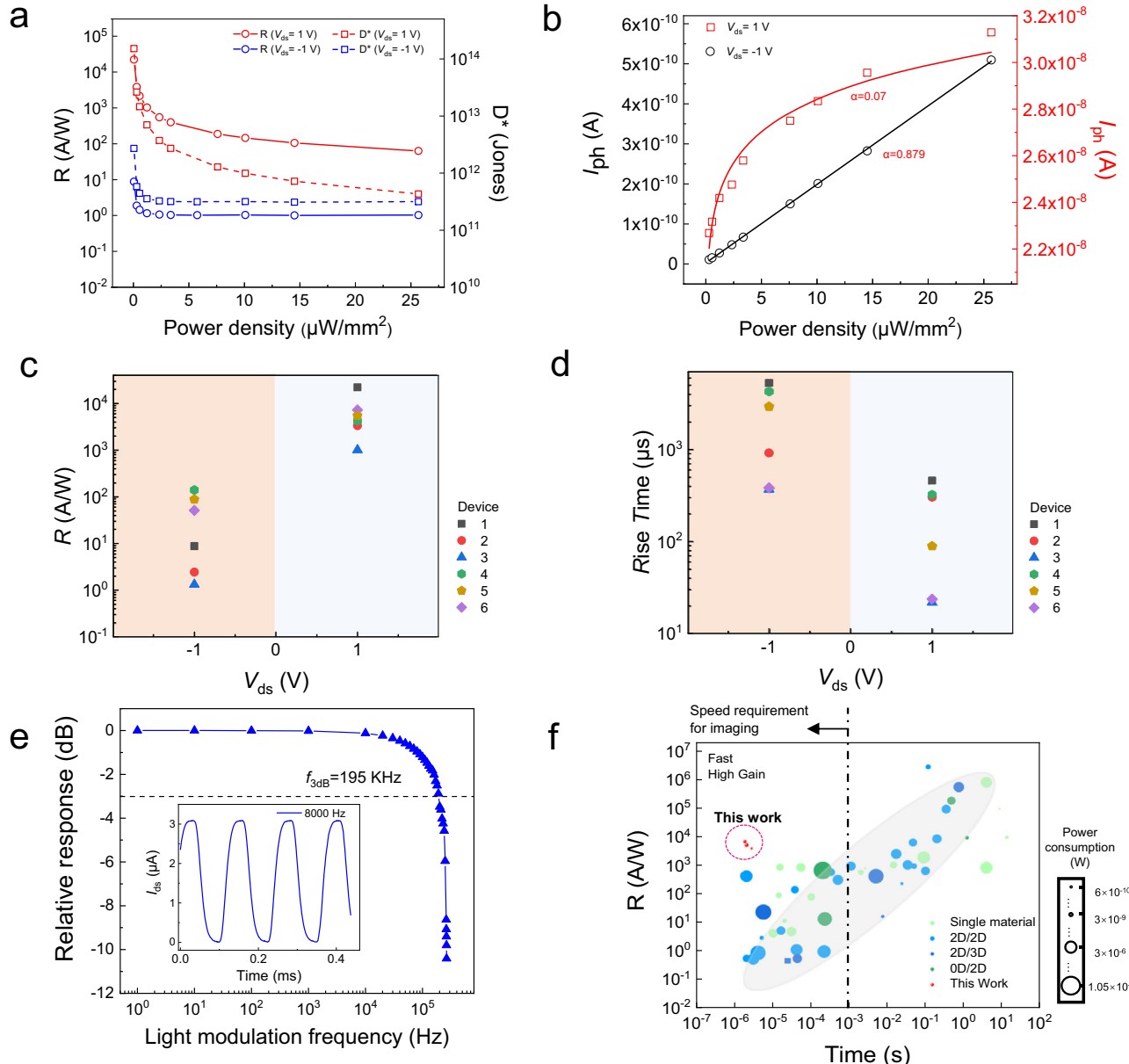

**Fig. 3 | The responsivity-speed relations of the WSe$_2$/Ta$_2$NiSe$_5$ heterostructure.**
**a** Extracted device responsivities, $R$ (solid lines), and detectivities, $D^*$ (dashed lines), at $V_{ds}=-1$ V and 1 V, respectively. **b** Extracted power dependence of the measured photocurrent at different $V_{ds}$. The solid lines are the exponential fitting between photocurrent and optical power. **c** Responsivity distribution of devices with different thicknesses. WSe$_2$/Ta$_2$NiSe$_5$: Device 1(D1): 60/13 nm; D2: 69/17 nm; D3: 35/23 nm; D4: 13.5/32.9 nm; D5: 10.1/11.6 nm; D6: 6.9/24.1 nm. The different shaded area indicates positive and negative bias conditions. **d** Response time of devices with different thicknesses. **e** Relative response with the modulation frequency

measured for Device 6. Inset: The time-dependent current of the device. The $f_{3dB}$ is defined as the laser modulation frequency when the photocurrent reduces to 0.707 of the maximum value. **f** The performance comparison with previous reports (gray ellipsoid). Details of quoted references should refer to Supplementary Fig. 18. The WSe$_2$/Ta$_2$NiSe$_5$ heterostructures with different thicknesses are included. Among the devices that fulfill the 1 ms speed limit (The vertical dashed line) for imaging applications, our work shows both high responsivity, fast response and low power consumption.

derived from the time-resolved dark currents (Supplementary Note 11), and the corresponding specific detectivity results display similar trends as $R$ (Fig. 3a). In addition, by fitting the power-dependent photocurrents with a power-law relationship, $I_{ph} \propto P^\alpha$ (Fig. 3b), the $\alpha$ is fitted to be 0.879 at $V_{ds}=-1$ V, in stark contrast, the $\alpha$ is 0.07 at $V_{ds}=+1$ V, which means more traps or recombination centers participate in the photoresponse under positive $V_{ds}$ than negative $V_{ds}$[26,58]. According to previous reports, there is a trap-induced trade-off between responsivity and response speed (if photogating effect plays a dominant role)[26,46]. However, in our device, this challenge is overcome with significant bias regulation, i.e., not only does the device

responsivity improves by three orders of magnitude when +1 V bias is applied compared to −1 V bias, the response speed also improves by an order of magnitude (Fig. 3c, d), simultaneously. These phenomena reveal that the speed constrains in defects-induced photogating effect is mitigated at positive $V_{ds}$. Similar phenomena were also found under the illumination of other wavelengths (e.g., 532 and 635 nm in Supplementary Note 10). Notably, the 3 dB cutoff frequency measured for the devices can reach a high value of ~195 kHz at $V_{ds}=1$ V (Fig. 3e). The response time of the device is estimated to be ~1.8 μs by the equation: $f_{3dB}=0.35/t_r$, where $t_r$ is the response time of the device, and the corresponding responsivity is 7.3 ×10$^3$ A/W at $V_{ds}=1$ V (details shown in

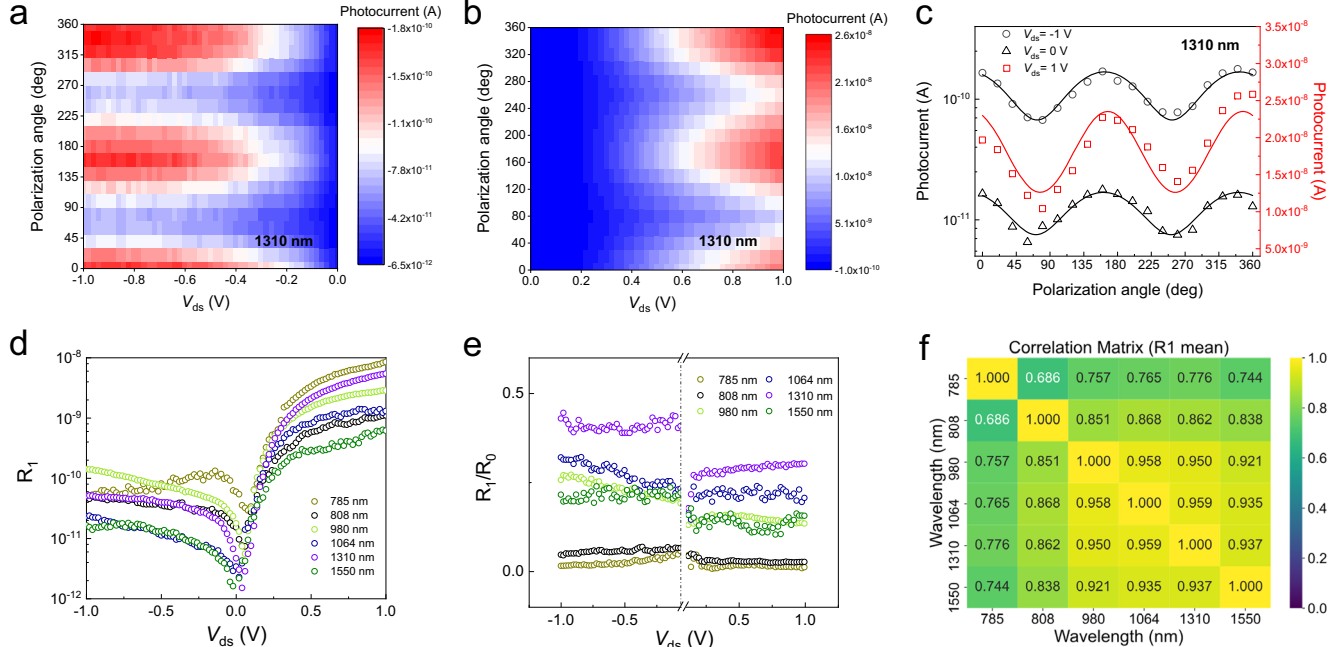

**Fig. 4 | The photogating-empowered polarization photoresponse and wavelength discrimination in the WSe₂/Ta₂NiSe₅ heterostructure. a, b** Color plots of the polarization photocurrent under 1310 nm illumination at various bias voltages $V_{ds}$. The photoresponse of WSe₂/Ta₂NiSe₅ photodetector can be tuned by the $V_{ds}$ and the incident light polarization angle, $\theta$. **c** The polarization photocurrents under 1310 nm illumination. The photocurrent showcases the same dependency on polarization angle, $\theta$, as sweeping $V_{ds}$ from −1 V to 1 V. The solid lines are fits to the polarized photocurrent. **d** The extracted polarization sensitive contributions $R_1$

($V_{ds}$, $\lambda$) from the polarization photocurrents. $R_1$ depends on both the bias voltage $V_{ds}$ and the incident light wavelength. **e** The anisotropic ratio $\beta$, $R_1/R_0$, extracted from the polarization insensitive ($R_0$) and sensitive ($R_1$) contributions of the polarization photocurrent. The vertical dashed line indicates the two sectors being analyzed under positive and negative bias conditions. **f** The calculated correlation matrix between different wavelengths, with the element as the correlation coefficient of two columns of $R_1$: corr{$R_1(V_{ds}, \lambda = \lambda_1)$, $R_1(V_{ds}, \lambda = \lambda_2)$}, where $\lambda_1$ and $\lambda_2$ are two different wavelengths.

Supplementary Note 9). The bias tunability of our device provides a facile way to boost the device performance, and a target region that can overcome the responsivity-speed limit was reached with small power consumption (with measured values from 0.762 nW to 13 nW) which is comparable to or even lower than the commercial photodetectors based on silicon, germanium and indium gallium arsenide (e.g. FDS1010, FD10D, DSD2 from Thorlabs), as illustrated in Fig. 3f (see more details in Supplementary Fig. 18).

**Photogating-empowered polarized response under positive bias**

Since the number of the photoexcited-carriers in the Ta₂NiSe₅ is sensitive to polarization of incident light due to its in-plane anisotropic crystal structure, thereby, the light induced photogating at positive $V_{ds}$ is polarization sensitive as well. The polarization-sensitive photogating will empower the device polarized light photodetection capability under positive $V_{ds}$ even though the photocurrent is mainly generated at the metal/WSe₂ Schottky junction. Besides, the relatively large bandgap of WSe₂ (~1.2 eV) normally limits the device photoresponse to below ~1000 nm. Nevertheless, in the above polarization-sensitive photogating process, the absorption of small-bandgap Ta₂NiSe₅ determines the operating wavelength range, which will surpass the limitation of photoresponse range of WSe₂ itself.

To verify the above analysis, we characterized our device at different illumination wavelengths in the infrared range up to 2200 nm (Supplementary Note 12). Considering the small bandgap of Ta₂NiSe₅ (~0.3 eV), our device should also work at even longer wavelengths. The peak EQE reaches $3.5 \times 10^6$ % at 785 nm illumination and $V_{ds} = 1$ V (Supplementary Note 13), and the EQE decreases to 157% at 2200 nm wavelength.

In the following, we investigated the device polarization photoresponse under light illumination of different wavelengths. The schematic light path and measurement results are depicted in

Supplementary Fig. 22. The polarization angle $\theta$ is defined as the angle between light polarization and the metal electrode edge. The photocurrent varied periodically when the $\theta$ is rotated counterclockwise from 0° to 360°, and reached the maximum when light polarization is parallel to the elongated axis of Ta₂NiSe₅, which is corresponding to the $a$-axis of the flake. Further measurements reveal that the polarization photocurrent showcases the same dependency on crystal orientation as sweeping $V_{ds}$ from −1 V to 1 V (Fig. 4a–c). The polarization dependent photocurrent can be described by a general formula: $I_{ph} = R_0 + R_1 \cdot \cos(\theta + \theta_0)$, with $R_0$ and $R_1$ representing the polarization insensitive and sensitive contributions, respectively. Once the two factors have been extracted, we can calculate the two widely used figure of merits in polarization photodetectors: the polarization ratio as $|(R_0 + R_1)/(R_0 - R_1)|$ and the anisotropic ratio $\beta$ as $R_1/R_0$. Notably, both the $R_0$ and $R_1$ are functions of applied source-drain bias and the wavelength of incident light, $R_0 = R_0 (V_{ds}, \lambda)$ and $R_1 = R_1 (V_{ds}, \lambda)$, as shown in Fig. 4d, e. The polarization dependence is a result of the anisotropic structure of Ta₂NiSe₅, in which the optical absorption coefficient along $a$ axis is calculated to be higher than that along $c$ axis for both mono- and multilayer Ta₂NiSe₅ (Supplementary Fig. 23). The calculated ratio of absorption coefficient along different crystalline orientations ($a/c$) varies for different wavelengths and reaches the peak value around 1310 nm, accounting for the wavelength dependent $R_0$ and $R_1$, in agreement with our experiments. Furthermore, the applied source-drain bias could alter the scattering of the photocarriers in the photogating process of our device so that the anisotropic response also becomes bias-tunable (Supplementary Note 14). The dependence of photoresponse on both bias and wavelength allows our device to discriminate different wavelengths by sweeping the bias. Figure 4f shows the calculated correlation matrix between different wavelengths, with the element as the correlation coefficient of two columns of $R_1$: corr{$R_1(V_{ds} = \lambda_1)$, $R_1(V_{ds} = \lambda_2)$}, where $\lambda_1$ and $\lambda_2$

are two different wavelengths. It is worth noting that our method only requires two electrodes, a simple configuration for practical implementation of high-resolution pixels.

## Discussion

In summary, we report a WSe$_2$/Ta$_2$NiSe$_5$ heterostructure with simultaneously improved responsivity and speed, with the aid of photogating-assisted carrier tunneling. The proposed mechanism helps achieve orders of magnitude faster carrier transit time, and shortened carrier lifetime by managing the trap sites, providing a state-of-the-art approach to address the trade-off between responsivity and speed in photodetectors based on photogating effect. Notably, the metal-induced gap state, external disorders/defects induced gap state or interface dipoles may lead to Fermi level pinning (FLP) effect at the metal contact interface[59]. If fully de-pining is realized with van der Waals contacts[60], the device photoresponse could be further regulated by the gating effect. The observed wavelength-dependent polarization characteristics under positive biases validate that the photogating is polarization-sensitive. In the future work, the polarization photodetection in the infrared region could be further improved by coupling with plasmonic nanostructures or integrating with a circuitry amplification system, and we believe this has practical application for highly sensitive detection in multiple fields. Importantly, the tunable photoresponse of our device by switching source-drain voltages give us insight into the spectral information of incident lights, from which it is possible to reconstruct spectra from their corresponding photoresponse vectors by a unique spectral learning procedure in the future[61–63]. Our results combine the physics of addressing the responsivity-speed trade-off and broadband polarized photodetection with wavelength discrimination, providing possibilities for exploring novel on-chip optoelectronic applications, such as in polarization imaging, high contrast polarizer, miniaturized spectrometer, etc.

## Methods

### Device fabrication

Ta$_2$NiSe$_5$ and WSe$_2$ nanosheets were fabricated via the mechanical exfoliation method from single crystals grown by the chemical vapor transport (CVT) method[44]. Firstly, thin flakes of Ta$_2$NiSe$_5$ were exfoliated from its bulk crystals using scotch tape and then transferred onto a silicon substrate (with 300 nm SiO$_2$). Then, multilayer WSe$_2$ were exfoliated onto the polydimethylsiloxane (PDMS) film and transferred selectively on top of the Ta$_2$NiSe$_5$ flake under the optical microscope assisted by an 3D positioning system. To fabricate the device for measurements, the source/drain electrodes were patterned by ultraviolet photolithography, and Ti/Au (10/80 nm) metals were deposited by the thermal evaporation.

### Characterization of WSe$_2$/Ta$_2$NiSe$_5$ heterostructure

The morphologies of the WSe$_2$/Ta$_2$NiSe$_5$ heterostructure were investigated by an optical microscope (BX51, OLMPUS). The Raman mapping and PL mapping were carried out at room temperature by a confocal Raman/PL system (Alpha 300R, WITec) equipped with 532 and 633 nm laser sources. The thicknesses of the WSe$_2$ and Ta$_2$NiSe$_5$ nanoflakes were measured using atomic force microscopy (Cypher S, Asylum Research). The absorption spectra of the materials were measured using a customized microfocused absorption system.

### Optoelectrical measurements

The electrical measurements were performed under ambient conditions at room temperature. All static behaviors of the photodetector were characterized by a semiconductor parameter analyzer (Keithley 4200) on a probe station (EVERBEING, C-4) in the dark and under illumination by different lasers: IR (2200, 1550, 1310, 1064, 980 and 808 nm), red (635 nm), green (532 nm). The device has been measured multiple times to ensure the consistency of the dark current, and the

spot area of incident light was confirmed based on an optical microscope[64]. The temporal responses of the device were recorded by a current meter after the light illumination switching on-off. The device 3 dB bandwidth is measured by modulating the laser switching frequency through a signal generator (RIGOL, DG822), and the modulated optical signal was focused on the photodetector through an optical microscope.

### Photocurrent mapping

The spatial-resolved photocurrent mapping was conducted using scanning photocurrent microscopy built on a confocal Raman/PL system (WITec, Alpha 300R) with a high spatial resolution of about 350 nm. The device was laterally moved with steps of 0.5 μm, where a focused laser beam (532/633 nm) was raster-scanned over the whole device area. The source-drain current $I_{ds}$ was recorded by a current meter under various bias voltages $V_{ds}$.

### Polarization-sensitive characterization

A linear polarizer (Thorlabs, LPVIS050) and half-wave plate were used to generate polarized light impinging on the sample in order to measure the polarization-dependent characteristics. The polarization angle was changed by rotating the polarizer.

### DFT calculations

All calculations were performed with the Vienna ab initio simulation package[65]. The projector augmented-wave (PAW) method was applied and the van der Waals interactions were considered by using DFT-D3 method[66]. For the geometry optimization and electronic structure calculation with Perdew–Burke–Ernzerhof (PBE) functionals[67], the plane-wave cutoff energy is 500 eV and the k-point mesh for the first Brillouin zone is $4 \times 10 \times 1$ for both Ta$_2$NiSe$_5$ and WSe$_2$/Ta$_2$NiSe$_5$ heterostructures. A vacuum space of 20 Å is set to avoid the interaction between the adjacent slabs. The convergence of force and total energy is 0.01 eV Å$^{-1}$ and $10^{-7}$ eV, respectively.

The optical absorption coefficient of Ta$_2$NiSe$_5$ is calculated by the following equation[68]:

$$\alpha(\omega) = \sqrt{(2)}\,\omega \left[ \sqrt{\varepsilon_1^2(\omega) + \varepsilon_2^2(\omega)} - \varepsilon_1(\omega) \right]^{1/2} \tag{1}$$

where $\varepsilon_1(\omega)$ and $\varepsilon_2(\omega)$ are the real and imaginary parts of the complex dielectric function, respectively. $\varepsilon_1(\omega)$ can be obtained from $\varepsilon_2(\omega)$ based on the Kramer–Kronig relationship.

## Data availability

All technical details for producing the figures are enclosed in the supplementary information. Data are available from the corresponding authors D.L., C.-W.Q. or S.L. upon request.

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

## Acknowledgements

S.L. acknowledges the financial support from the National Natural Science Foundation of China (No. 62121005, 62022081, 62334010 and 61974099), the National Key Research and Development Program (2021YFA0717600), the Natural Science Foundation of Jilin Province (20210101173JC) and Changchun Key Research and Development Program (21ZY03). K.S.N. is grateful to the Ministry of Education, Singapore (Research Centre of Excellence award to the Institute for Functional Intelligent Materials, I-FIM, project No. EDUNC-33–18-279-V12) and Royal Society (UK, grant 304 number RSRP\R\190000) for support. C.-W.Q. acknowledges financial support from the National Research Foundation (Grant No. NRF-CRP26-2021-0004).

## Author contributions

S.L., C.-W.Q. and D.L. conceived of the original concept and supervised the project. M.L., J.W. and S.L. performed most of the experiments. L.Q. and Z.S. contributed to the computational theoretical analysis. M.L. and Y.L. contributed to material preparations and characterizations. M.L. and Y.L. contributed to the photoresponse measurements. M.L., J.W., L.Q., J.A., X.L., Y.L., Z.S., K.S.N., D.L. and S.L. analyzed the data and co-wrote the paper. All authors discussed the results and commented on the manuscript. All authors have approved the final version of the manuscript.

## Competing interests

The authors declare no competing interests.
