## [Peer Review File · Nature Communications]

Photogating-Assisted Tunneling Boosts the Responsivity and Speed of Heterogeneous WSe₂/Ta₂NiSe₅ PhotodetectorsEditorial Note: Parts of this Peer Review File have been redacted as indicated to remove third-party material where no permission to publish could be obtained.

REVIEWER COMMENTS

Reviewer #1 (Remarks to the Author):

The reported results of the manuscript are fully convincing. The studied photogating-assisted tunneling effect in WSe₂/Ta₂NiSe₅ heterostructure detectors generated high responsibility together with high speed. These two properties have been difficult to achieve simultaneously in earlier reports.

The findings are based on numerous samples, their multiple measurements and also modelling efforts. The selected methods are not just many, but they are used to get a full picture of the studied materials and their properties in the heterostructure.

The chosen combination of 2D materials provides in addition to the properties already shown in the title, but also polarization detection and possibility for multi-color sensing. These results are thoroughly discussed in the manuscript and in Supplement.

I found very few open questions or minor flaws:

1. Ta₂NiSe₅ has been shown to exhibit layered monoclinic structures below the transition temperature (328 K). What will happen above that and, especially, how does it change the band structure? I did not find this in the references (39-41).
2. This sentence is not clear to me: "a large number of traps originate from photogenerated electrons in the Ta₂NiSe₅ conduction band that are blocked by the energy band potential under positive bias, which acts as shallow trap site with a shorter lifetime". Please rephrase this.

Reviewer #2 (Remarks to the Author):

Comments:

The manuscript by Mingxiu Liu, et al. has comprehensively investigated a photodetector with photogating-assisted tunneling for responsivity and speed improvement composed of van der Waals WSe₂/Ta₂NiSe₅ heterostructure configuration, enhanced simultaneously, overcoming the conventional trade-off between responsivity and speed. The photodetection performance in both visible and NIR ranges has been measured, under electrically gated tuning. The polarization detection was also addressed, providing possibilities for exploring novel on-chip optoelectronic applications. Although the authors have provided very interesting experimental results, there are a few unclear claims and I also doubt some figure-of-merits they have provided. The authors should provide more details and revise the manuscript thoroughly before it can be considered for publication.

#1. Is there any evidence of the Fermi pinning effect in Figs. 2 (c) and (d). If so, they shall elaborate more

on its origins and impacts on device performance. What is the influence of the pinning for the band alignments and their photodetection performance?

#2. Any optical absorptance and/or reflectance spectra for individual layers? Any optical absorption and/or reflection spectra for individual layers in combination with the heterostructures?

#3. Please explain the mechanism for EQE peak value of 3.5×10^6 % at 785 nm illumination. Also, please clarify the mechanism for the EQE of 157 % at 2200 nm.

#4. To clarify the tunneling process, it is suggested that a simple variable-temperature-dependent I-V curve in Figure 2 is provided. By considering that the measurement of the photo current was at $V_g=0$, the authors should provide the characterization and calculation details. Also, they should provide the measurement of the dark current was at $V_g=0$ in Fig. 1(c).

#5. The authors should pay attention to how to correctly characterize figures of merit of two-dimensional photodetectors. A useful reference is found here

F. Wang, T. Zhang, W. D. Hu, et al., Nature Communications 14 (1), Apr 19 (2023)
(Nature Communications, 2023, DOI:10.1038/s41467-023-37635-1)

<https://www.nature.com/articles/s41467-023-37635-1>

#6. A relevant reference may be useful for discussion and citation, as shown below.

L. Tao, et al., npj 2D Materials and Applications 1, 19823–829 (2017).

<https://www.nature.com/articles/s41699-017-0016-4>

Reviewer #3 (Remarks to the Author):

This manuscript reported a WSe₂/Ta₂NiSe₅ heterostructure detector. The photodetection gain and response speed can be enhanced by the photogating effect and direct tunneling, overcoming the tradeoff between responsivity and speed. However, there are some serious problems in the manuscript. Mainly, the photoelectric mechanism and the performance are not novel and good for two-dimensional photodetectors. This manuscript does not meet the requirement of Nature Communications. Thus, I do not recommend publishing it on Nature Communications. The comments are as follows.

1. There are some studies about the tradeoff between responsivity and speed for photodetectors (npj 2D Mater Appl, 2017, 1, 19; Adv. Electron. Mater. 2021, 7, 2000920; ACS Appl. Mater. Interfaces 2018, 10, 36512–36522). For graphene/MoS₂/Si, the responsivity and response time are 3×10^4 A/W and 500 ns, respectively. These parameters are better than those of the WSe₂/Ta₂NiSe₅ heterostructure. Moreover, the graphene/MoS₂/Si operates without the gate voltage. The authors demonstrated that photogenerated carriers are accumulated and tunneled through MoS₂, which is similar to that of WSe₂/Ta₂NiSe₅ photodetectors. Thus, I could not find the innovation of WSe₂/Ta₂NiSe₅ photodetectors, compared to graphene/MoS₂/Si detectors.

2. For energy band diagrams of the WSe₂/Ta₂NiSe₅ heterostructure, the projected band structures by density functional theory (DFT) methods are not enough. The energy differences of WSe₂ and Ta₂NiSe₅ between Fermi energy level and conduction band minimum are very key for energy band alignment.

3. In the manuscript, the authors said that “the Fermi level of WSe₂ will further downshift and the hole concentration is increased, enabling the direct tunneling of the holes from the drain towards the source

terminal". Please provide more explanations. How does the direct tunneling appear between WSe₂ and Ta₂NiSe₅?

Manuscript ID: NCOMMS-23-33817

Title: Photogating-Assisted Tunneling Boosts the Responsivity and Speed in a Heterogeneous Photodetector

Point-by-point responses to Reviewers' Comments

We are very grateful for the insightful comments from the reviewers, which are valuable for further improving the quality of this manuscript. We have tried our best to address all the concerns raised by the reviewers, and the point-by-point responses to the reviewer's comments are listed below. The changes to the manuscript are indicated in **RED** in the marked version of the revised manuscript. We hope that the quality of our manuscript has met the publication standard after this revision.

Reviewer #1

The reported results of the manuscript are fully convincing. The studied photogating-assisted tunneling effect in WSe_2/Ta_2NiSe_5 heterostructure detectors generated high responsibility together with high speed. These two properties have been difficult to achieve simultaneously in earlier reports. The findings are based on numerous samples, their multiple measurements and also modelling efforts. The selected methods are not just many, but they are used to get a full picture of the studied materials and their properties in the heterostructure. The chosen combination of 2D materials provides in addition to the properties already shown in the title, but also polarization detection and possibility for multi-color sensing. These results are thoroughly discussed in the manuscript and in Supplement.

I found very few open questions or minor flaws:

[Reply] We sincerely thank the reviewer's positive comments on our manuscript such as "*fully convincing*". We have carefully addressed the concerns raised by the reviewers, as shown below.

[Comments 1]: *Ta₂NiSe₅ has been shown to exhibit layered monoclinic structures below the transition temperature (328 K). What will happen above that and, especially, how does it change the band structure? I did not find this in the references (39-41).*

[Reply]: We thank the reviewer for this comment. Above the transition temperature (328 K), Ta₂NiSe₅ transforms from monoclinic phase to orthorhombic phase with small shearlike lattice distortion (see below **Figure R1a, b**). In addition, the band gap of monoclinic Ta₂NiSe₅ is 0.36 eV induced by the band hybridizations due to the low symmetry (ACS Appl. Mater. Interfaces 2021, 13, 17948), while the band gap of orthorhombic Ta₂NiSe₅ is closed (semi-metallic) due to the high symmetry in orthorhombic phase. The difference in band structures of monoclinic and orthorhombic phase of Ta₂NiSe₅ is shown in below **Figure R1c, d** (Phys. Rev. Res. 2020, 2, 013236).

[REDACTED]

Figure R1. (a), (b) The atomic structures of Ta₂NiSe₅ in the orthorhombic and monoclinic phases, respectively. (c), (d) The band structures of Ta₂NiSe₅ in the orthorhombic and monoclinic phases [Phys. Rev. Res. 2020, 2, 013236].

It should be pointed out that **the orthorhombic phase of Ta₂NiSe₅ cannot be maintained at room temperature**. Besides, we have observed polarization sensitive photoresponse throughout our measurements, so we conclude our materials is in monoclinic phase rather than orthorhombic phase. The photoresponse of orthorhombic phase Ta₂NiSe₅ would be interesting for future studies but is out of the scope of this

manuscript.

[Actions]: On page 5, we have updated the corresponding references as Ref. 36 and Ref. 37 (ACS Appl. Mater. Interfaces 2021, 13, 17948; Phys. Rev. Res. 2020, 2, 013236) in the revised version of the manuscript.

[Comments 2]: *This sentence is not clear to me: "a large number of traps originate from photogenerated electrons in the Ta₂NiSe₅ conduction band that are blocked by the energy band potential under positive bias, which acts as shallow trap site with a shorter lifetime". Please rephrase this.*

[Reply]: We thank the reviewer for this comment. We have followed the reviewer's suggestion and rephrased this sentence in our revised main text.

On Page 10 of the revised main text, we have rephrased this sentence to “**a large number of photogenerated electrons are blocked in the Ta₂NiSe₅ conduction band by the barrier of energy band under positive bias, which act as the role of shallow trap sites that capture photogenerated carriers with a shorter lifetime.**”

We would like to provide here more detailed explanations of this sentence: Under positive bias, the photo-generated electrons in the conduction band of Ta₂NiSe₅ are blocked by the potential barriers at the heterojunction interface, resulting in photogating effect. The resulted photogating effect further regulates the hole concentration in the device, achieving a significant improvement in responsivity. Notably, the blocked electrons can be released flexibly as switching the external bias, from this prospective, it behaves like the role of shallow trap sites that capture photogenerated carriers with a shorter lifetime.

Reviewer #2

The manuscript by Mingxiu Liu, et al. has comprehensively investigated a photodetector with photogating-assisted tunneling for responsivity and speed improvement composed of van der Waals WSe₂/Ta₂NiSe₅ heterostructure configuration, enhanced simultaneously, overcoming the conventional trade-off between responsivity and speed. The photodetection performance in both visible and NIR ranges has been measured, under electrically gated tunneling. The polarization detection was also addressed, providing possibilities for exploring novel on-chip optoelectronic applications. Although the authors have provided very interesting experimental results, there are a few unclear claims and I also doubt some figure-of-merits they have provided. The authors should provide more details and revise the manuscript thoroughly before it can be considered for publication.

[Reply]: We sincerely thank the reviewer's positive comments on our manuscript such as "comprehensively investigated" and "very interesting experimental results". Meantime, we highly appreciate the reviewer for raising concerns on the mechanism, characterization, and literatures. We have carefully addressed the concerns raised by the reviewers, as shown below.

[Comments 1]: *Is there any evidence of the Fermi pinning effect in Figs. 2 (c) and (d). If so, they shall elaborate more on its origins and impacts on device performance. What is the influence of the pinning for the band alignments and their photodetection performance?*

[Reply]: We thank the reviewer for the constructive comment. The Fermi level pinning (FLP) effect at the metal-semiconductor contact is expected to exist in our device. We would like to further elaborate on this as below.

The contact property between 2D materials and metal electrodes has important impact on device performance. According to current studies, the metal-induced gap state (MIGS), external disorders/defects induced gap state (DIGS) or interface dipoles

lead to the FLP effect with high contact resistance, and lowered device mobility etc. (Nat. Mater. 2015, 14, 1195; Sci. Adv. 2016, 2, e1600069). For our device, we directly deposit Ti/Au metal on the 2D material, which will inevitably introduce defects and DIGS at the contact interface (Nat. Commun. 2023, 14, 111; Adv. Mater. 2022, 34, 2108425; Adv. Funct. Mater. 2022, 32, 2204288), thereby FLP at the metal-semiconductor contact is expected to exist.

We next turn to the impact of the FLP effect in our device. Based on the photocurrent mapping results in our work, the maximum photocurrent under positive bias is generated near the junction region where the metal electrode contacts WSe₂ (see below **Figure R2b**). Thereby, we focus on analyzing the effect of FLP at the metal/WSe₂ contact interface. The existence of the FLP effect in the contact between WSe₂ and the metal electrode has been investigated previously (Nat. Commun. 2020, 11, 1866; Nat. Electron. 2022, 5, 241; Nat. Electron. 2021, 4, 342). Although the Schottky barrier at metal/2D materials interface is difficult to modulate with the change of external bias voltage, the band alignment inside our device is bias-tunable as can be observed from the photocurrent mapping measurements results (see below **Figure R2**). The photocurrent generation positions of the device are different under positive and negative biases, indicating the bias adjustability of the band alignment of the device. In addition, the device has good gate control capabilities, as proved by the on/off ratio of 10⁴ (see below **Figure R3**), which further demonstrates the tunability of Fermi level inside the device channel.

Furthermore, if fully de-pining (for instance with van der Waals contacts) is realized in our device, the Fermi level of WSe₂ is anticipated to be more susceptible to the gating effect. Accordingly, the photogating effect may play a stronger effect on modulating the concentration of the photogenerated carriers. In that case, the gain and responsivity of the device could be further regulated.

Figure R2 a-b, The spatially resolved photocurrent mapping images at (a) $V_{ds} = -1$ V and (b) $V_{ds} = 1$ V.

Figure R3 Transfer curves (I_{ds} - V_{gs}) of the WSe₂ device at $V_{ds} = 1$ V.

[Actions]:

On Page 15 of the revised manuscript, we have added below descriptions:

“Notably, the metal-induced gap state, external disorders/defects induced gap state or interface dipoles may lead to Fermi level pinning (FLP) effect at the metal contact interface.⁵⁹ If fully de-pinning is realized with van der Waals contacts,⁶⁰ the device photoresponse could be further regulated by the gating effect.”

[Comments 2]: Any optical absorptance and/or reflectance spectra for individual layers? Any optical absorption and/or reflection spectra for individual layers in

combination with the heterostructures?

[Reply]: We thank the reviewer for raising concerns on the absorption data. We have measured the optical absorption of WSe₂, Ta₂NiSe₅ and WSe₂/Ta₂NiSe₅ heterojunction on sapphire substrates (see below **Figure R4**). The results show that the absorption of the heterojunction in visible light range is contributed by WSe₂ and Ta₂NiSe₅, while the near-infrared photogenerated carriers are mainly provided by optical absorption of Ta₂NiSe₅.

Figure R4 UV-vis-NIR absorption spectra of Ta₂NiSe₅, WSe₂ and WSe₂/Ta₂NiSe₅.

[Actions]:

On Page 5 of the revised Supplementary Information, we have added the absorption spectrum of the device as shown in the revised **Supplementary Fig. 3f**.

Supplementary Fig. 3 Optical characterization of the WSe₂/Ta₂NiSe₅

heterojunction. a, The non-polarized Raman spectra collected from isolated WSe₂, Ta₂NiSe₅, and the overlapped heterojunction region. **b-c**, The corresponding Raman mapping images measured at the characteristic Raman peaks of (b) Ta₂NiSe₅ (121 cm⁻¹) and (c) WSe₂ (260 cm⁻¹). **d**, PL spectra of isolated WSe₂, Ta₂NiSe₅, and the WSe₂/Ta₂NiSe₅ heterojunction. **e**, The corresponding PL mapping image of the WSe₂/Ta₂NiSe₅ heterojunction. **f**, **UV-vis-NIR absorption spectra of Ta₂NiSe₅, WSe₂ and WSe₂/Ta₂NiSe₅.**

[Comments 3]: Please explain the mechanism for EQE peak value of 3.5×10^6 % at 785 nm illumination. Also, please clarify the mechanism for the EQE of 157 % at 2200 nm.

[Reply]: We sincerely thank the reviewer for raising concerns on the mechanism of the measured high EQE.

The external quantum efficiency (*EQE*) can be expressed as: $EQE = I_{ph}/P_{in} (\hbar c/e\lambda) = R (\hbar c/e\lambda)$ (Nat. Commun. 2023, 14, 2224; Nat. Commun. 2019, 10, 4663), the *EQE* is closely related to responsivity *R* or optical gain, and the incident wavelength λ . To explain the mechanism for the high *EQE*, we would like to further elaborate on this point as below.

First, for the high EQE peak value at 785 nm illumination, the absorption of the heterojunction in this range is mainly dominated by the resonance absorption of WSe₂, as shown in above **Figure R4**. In particular, the strong exciton absorption of the WSe₂ around 785 nm leads to a huge increase in photogenerated carriers (Nat. Commun. 2017, 8, 1296; Adv. Mater. 2022, 34, 2108412). Notably, based on the photogating-assisted tunnelling mechanism proposed in our work, a large number of photogenerated electrons are blocked by the potential barrier at the heterostructure interface under positive bias, which act as the role of trap sites that capture photogenerated electrons and finally leads to a high optical gain (photogating effect). Therefore, our device shows a high *EQE* peak value at 785 nm illumination.

Second, for the *EQE* of 157 % at 2200 nm, based on the absorption spectrum, the

absorption of the device under infrared light mainly arises from the Ta₂NiSe₅. It should be noted that the light absorption in the infrared spectrum is much lower than that of in visible light range (**Figure R4**), which limits the external quantum efficiency in the near infrared region. Besides, due to the photogating effect in our device as claimed above, the *EQE* of our device at 2200 nm is still greater than 100%. Overall, these results indicate that our device has good photoelectric conversion efficiency across the visible to near-infrared spectrum.

Figure R4 UV-vis–NIR absorption spectra of Ta₂NiSe₅, WSe₂ and WSe₂/Ta₂NiSe₅.

[Comments 4]: *To clarify the tunneling process, it is suggested that a simple variable-temperature-dependent I - V curve in Figure 2 is provided. By considering that the measurement of the photocurrent was at $V_g=0$, the authors should provide the characterization and calculation details. Also, they should provide the measurement of the dark current was at $V_g=0$ in Fig. 1(c).*

[Reply]: We sincerely thank the reviewer for his/her constructive advices

First, we have followed the reviewer’s suggestion and conducted the temperature-dependent I - V characteristics measurements, as shown in below **Figure R5a, b**. As can be observed, the plot of $\ln(I/V^2)$ versus I/V displays small variations as the temperature decreases from 300 K to 128 K, with the linear region of curves exhibiting nearly the same slope. The above results indicate that the output current of the device under forward bias has little dependence on temperature changes, which eliminates the

dominate impact of the carrier thermal excitation on device performance and further verifies the tunneling-dominated transport behavior of charge carriers in our device.

Figure R5 a, b Temperature-dependent transport characteristics of the device under light illumination. Incident light power: $0.47 \mu\text{W}/\text{mm}^2$. The incident wavelength is 785 nm.

Second, the measurement of the dark current and photocurrent has been set at $V_g=0$, in **Figure 1c** and **Figure 2e** of the main text in our previous submission. To make it clearer, we have added the measurement conditions in the revised figure (as seen below). We would like to further explain the characterization and calculation details of typical performance parameters including device responsivity, detectivity, and response speed, as below.

Figure 1c Source-drain I - V curves of a representative device in dark. No gate voltage was applied ($V_{gs}=0$ V). This device consists of an approximately 60-nm-thick WSe_2 flake on top of a 13-nm-thick Ta_2NiSe_5 flake.

Figure 2e Output curves (I_{ds} - V_{ds}) in dark and under illumination with different power densities. The incident light is at 532 nm wavelength, and $V_{gs} = 0$ V.

The responsivity of the photodetector is calculated according to the formula $R = I_{ph}/PS$, where I_{ph} is the photocurrent, P is the power density, and S is the effective working area of the device. According to above formula, a high responsivity is obtained with $R = 2.2 \times 10^4$ A/W (780 nm, $0.05 \mu\text{W}/\text{mm}^2$) when $V_{ds} = 1$ V is applied. The typical calculated device responsivity has been shown in **Figure 3a** of the main text (also shown below).

As the key figures of merit of photodetectors, the noise and specific detectivity are also characterized. To calculate these values, the time-resolved dark currents with applying a constant drain bias (V_{ds}) at -1 V, 0 V and 1 V, have been measured (as shown in below **Figure R6 a-c**). By taking the Fourier transform of dark current traces, the noise spectral densities (S_n) as a function of frequency are obtained as shown in **Figure R6 d-f**.

Theoretically, the noise current consists of several parts that can be expressed as:

$$\begin{aligned}
 i_{noise} &= \sqrt{i_{shot}^2 + i_{thermal}^2 + i_{1/f}^2 + i_{g-r}^2} \\
 &= \sqrt{2eI_d B + \frac{4KBT}{R_{\Omega}} + i(f, B)_{1/f}^2 + i(f, B)_{g-r}^2} \quad (1)
 \end{aligned}$$

where I_d is the dark current, e is the elementary charge, B is the bandwidth, T is

the temperature, and R_{Ω} is the shunt resistance of device (The value of R_{Ω} can be obtained by fitting the I - V experimental data under weak voltage). The $i_{thermal}$ is thermal noise, i_{shot} is shot noise, $i_{1/f}$ is $1/f$ noise, and i_{g-r} is generation-recombination noise, respectively. The $1/f$ noise and $g-r$ noise are dominant at low frequencies, ascribing to the interface traps or defects. As is shown in **Figure R6 d, e**, when a bias voltage of -1 V and 0 V are applied, since the noise spectrum is almost frequency independent at the bandwidth of 1 Hz, white noise is the primary source of noise current. Therefore, the noise current (i_{noise}) can be expressed as:

$$i_{noise} = \sqrt{2eI_d B + \frac{4KBT}{R_{\Omega}}} \quad (2)$$

where $\sqrt{2eI_d B}$ is short noise and $\sqrt{\frac{4KBT}{R_{\Omega}}}$ is thermal noise. Due to the ultra-large value of R_{Ω} in our device ($R_{\Omega} > 7.9 \text{ G}\Omega$), the $i_{thermal}^2$ is far less than i_{shot}^2 ($i_{thermal}^2 \ll (i_{shot}/10)^2$), so the noise current can be estimated as $i_{noise} = \sqrt{2eI_d B}$. It is obvious that the noise mainly comes from the dark current. According to the noise spectral density extracted at the bandwidth of 1 Hz, the calculated D^* of the device in the main text is 2.18×10^{12} Jones ($V_{ds} = -1 \text{ V}$) and 9.6×10^{10} Jones ($V_{ds} = 0 \text{ V}$) at 785 nm with a power density of $0.05 \text{ }\mu\text{W}/\text{mm}^2$, respectively.

However, under the bias voltage of +1 V, we found that the noise current has a weak frequency dependence at around the bandwidth of 1 Hz. Thus, the specific detectivity can be expressed as:

$$D^* = \frac{R\sqrt{A}}{S_n} \quad (3)$$

where R is responsivity and A is device active area. According to the formula (3), the D^* is 8.3×10^{13} Jones at 785 nm for the device with a power density of $0.05 \text{ }\mu\text{W}/\text{mm}^2$, which is very close to the detectivity (1.5×10^{14} Jones) calculated assuming the dark current dominates the noise current. The detailed analysis of the device noise and specific detectivity has been shown in **Supplementary Note 11**.

Figure 3a, Extracted device responsivities, R , and detectivities, D^* , at $V_{ds} = -1$ V and 1 V, respectively. The incident light is at 785 nm wavelength, and $V_{gs} = 0$ V.

Figure R6 Device noise and specific detectivity. **a-c**, The time-resolved dark currents at (a) $V_{ds} = -1$ V, (b) $V_{ds} = 0$ V and (c) $V_{ds} = 1$ V. **d-f**, The noise spectral densities (S_n) as a function of frequency at (d) $V_{ds} = -1$ V, (e) $V_{ds} = 0$ V and (f) $V_{ds} = 1$ V.

For the estimation of the device response speed, we used an oscilloscope to capture the time-dependent current variation curve as shown in below **Figure R7**, which demonstrates obvious rising and falling edges. In addition, in order to further estimate the response time, we also measured the electrical bandwidth of the photodetector. The 3 dB cutoff frequency measured for the devices can reach a high value of ~ 195 kHz at $V_{ds} = 1$ V. Accordingly, the response time of the device is estimated to be ~ 1.8 μ s.

Figure R7 a, The time-dependent current of the device. **b**, Relative response with the modulation frequency measured for the heterostructure in the main text under 785 nm light illumination.

[Actions]:

On Page 14 of the revised Supplementary Information, we have provided the above temperature-dependent I - V characteristics measurements results and relevant analysis as **Supplementary Note 7**.

On Page 23 of the revised Supplementary Information, the detailed analysis of the device noise and specific detectivity has been shown as **Supplementary Note 11**.

On Pages 19 and 20 of the revised Supplementary Information, we have added the analysis of the response speed as **Supplementary Fig. 15**.

“We then measured the relative response with optical modulation. The 3 dB bandwidth of devices are extracted according to the dependence of photocurrent on the optical modulation frequency. As shown in **Supplementary Fig. 15**, the 3 dB cutoff frequency measured for device 4-6 can reach up to 130-195 kHz under 785 nm illumination. The response time of the device is estimated to be 2.7 μs -1.8 μs by the equation: $f_{3\text{dB}} = 0.35/t_r$, where t_r is the response time of the device.”

Supplementary Fig. 15 a-c, Relative response with the modulation frequency measured for heterostructures with various thicknesses under 785 nm light illumination. **Inset:** The time-dependent current of the devices.

[Comments 5]: *The authors should pay attention to how to correctly characterize figures of merit of two-dimensional photodetectors. A useful reference is found here F. Wang, T. Zhang, W. D. Hu, et al., Nature Communications 14 (1), Apr 19 (2023) (Nature Communications, 2023, DOI:10.1038/s41467-023-37635-1) <https://www.nature.com/articles/s41467-023-37635-1>*

[Reply]: Thanks for the constructive comment.

We have carefully read this reference and checked again the general criteria for characterizing the quality factor of two-dimensional photodetectors from the aspects of device dark current, noise, responsivity, detectivity, etc. When we estimate the response performance of the photodetector, the data processing and evaluation methods generally coincide with the mentioned reference. We have evaluated the performance of our device from several parts that may be easily misestimated as mentioned in the above reference, and to make sure our results are rational. For instance, for the estimation of the dark current of the device, the photoelectric measurements we carried out is in a constant-temperature laboratory. During the experiments, we have measured multiple times to ensure the consistency of the dark current, and the device works at a low bias voltage (± 1 V). Thus, the estimation of the dark current of the device is reasonable. As for the device responsivity, it is suggested to use the photocurrent mapping method

mentioned in the above reference to extract the effective absorption area, the results are generally consistent with that we have calculated. In addition, the spot area of incident light was measured based on an optical microscope, to ensure that we did not overestimate the responsivity of the device. And in terms of the noise and detectivity of the device, we have calculated the detectivity of the device through the noise spectral power density measured by the device, which is in good agreement with the method described in the mentioned reference.

[Actions]:

On Page 17 of the revised main text, we have added the mentioned reference as **Ref. 64**.

“The device has been measured multiple times to ensure the consistency of the dark current, and the spot area of incident light was confirmed based on an optical microscope.⁶⁴”

[Comments 6]: *A relevant reference may be useful for discussion and citation, as shown below. L. Tao, et al., npj 2D Materials and Applications 1, 19823–829 (2017). <https://www.nature.com/articles/s41699-017-0016-4>*

[Reply]: Thanks for the suggestion. We have carefully read this reference, discussed and cited it as Ref. 53 in the revised manuscript. In the mentioned reference, by using MoS₂ as thin tunneling layer, the response speed of device is improved by orders of magnitude, which further confirms the rationality of the tunneling of carriers in the device to improve the response speed.

[Actions]:

On Page 9 of the revised main text, we have discussed and cited the above reference as Ref. 53.

“It is noteworthy that the tunneling of carriers have been proved to be feasible in improving the device response speed in recent reports.⁵³”

Reviewer #3

This manuscript reported a WSe₂/Ta₂NiSe₅ heterostructure detector. The photodetection gain and response speed can be enhanced by the photogating effect and direct tunneling, overcoming the tradeoff between responsivity and speed. However, there are some serious problems in the manuscript. Mainly, the photoelectric mechanism and the performance are not novel and good for two-dimensional photodetectors. This manuscript does not meet the requirement of Nature Communications. Thus, I do not recommend publishing it on Nature Communications. The comments are as follows.

[Reply]: We sincerely thank the reviewer's summary on our work as well as valuable critics on the novelty and performance, which help us to further improve the quality of this manuscript. We have carefully addressed the concerns point-to-point as below.

[Comments 1]: *There are some studies about the tradeoff between responsivity and speed for photodetectors (npj 2D Mater Appl, 2017, 1, 19; Adv. Electron. Mater. 2021, 7, 2000920; ACS Appl. Mater. Interfaces 2018, 10, 36512–36522). For graphene/MoS₂/Si, the responsivity and response time are 3×10000 A/W and 500 ns, respectively. These parameters are better than those of the WSe₂/Ta₂NiSe₅ heterostructure. Moreover, the graphene/MoS₂/Si operates without the gate voltage. The authors demonstrated that photogenerated carriers are accumulated and tunneled through MoS₂, which is similar to that of WSe₂/Ta₂NiSe₅ photodetectors. Thus, I could not find the innovation of WSe₂/Ta₂NiSe₅ photodetectors, compared to graphene/MoS₂/Si detectors.*

[Reply] Thank the reviewer for the comment. After carefully read the above studies, we would like to further clarify the innovation of our work from the following aspects.

First, **in the aspect of mechanism**, most of the current reports are based on either carrier tunneling effect or photogating effect. However, the carrier tunneling effect usually leads to the improvement in response speed while the responsivity changes very little (Nano Lett. 2017, 17, 453; Adv. Mater. 2020, 32, 1902039; Adv. Mater. 2021, 33,

2101449; Adv. Mater. 2021, 33, 2101243). While the photogating effect leads to improved responsivity while the response speed of the device may become several orders of magnitude slower (Nat. Nanotechnol. 2013, 8, 497; Nat. Nanotechnol. 2012, 7, 363; Adv. Funct. Mater. 2016, 26, 1938; Adv. Funct. Mater. 2017, 27, 1603605). The mechanism in article 1 as mentioned by the reviewer (npj 2D Mater. Appl. 2017, 1, 19) belongs to the carrier tunneling effect. It is mentioned that the tunnel layer only plays the role of reducing the density of interface defect states, which achieves an increase in response speed but not significantly improving the responsivity (within one order of magnitude). While the mechanism in article 3 (ACS Appl. Mater. Interfaces 2018, 10, 36512) belongs to the photogating effect. The photogating effect enables improved device responsivity (1 or 2 orders of magnitude), but the response speed is also inevitably reduced by several orders of magnitude.

It should be pointed out that the mechanism reported in this manuscript, namely **photogating-assisted tunneling**, is different from all the previous ones. This novel effect leverages the synergistic interplay between photocarrier multiplication and carrier acceleration through the tunnelling process. As demonstrated in our work, this effect allows simultaneous improvement both in responsivity (by about four orders) and response speed (by about one order), which largely overcome the traditional trade-off between responsivity and response speed.

Second, **in the aspect of device functionality**, the photogating effect in our device shows a dependence on the polarization states of incident light, which can be further tuned by source-drain voltages, allowing for wavelength discrimination with just a two-electrode planar structure. Therefore, the $\text{WSe}_2/\text{Ta}_2\text{NiSe}_5$ device in our work are no longer limited to the detection of light intensity, but have taken an important step towards the field of multi-functions including intensity, polarization and spectrum. Such a **versatile function** has been rarely reported in two-port devices as what we report here.

Third, it should be also noted that for article 2 mentioned by the reviewer (Adv. Electron. Mater. 2021, 7, 2000920), the reported performance parameters are based on theoretical prediction. The experimental verification of theoretical prediction remains

largely unexplored.

Based on all above, our work shows significant novelty in terms of new mechanisms and multiple-functions. Importantly, these characteristics help to achieve the long-sought next-generation photodetectors with high responsivity, fast speed, polarization detection, and multi-color sensing.

[Comments 2]: *For energy band diagrams of the WSe₂/Ta₂NiSe₅ heterostructure, the projected band structures by density functional theory (DFT) methods are not enough. The energy differences of WSe₂ and Ta₂NiSe₅ between Fermi energy level and conduction band minimum are very key for energy band alignment.*

[Reply] Thanks for the comment. We have carefully analyzed the energy band alignment of the heterostructure by combining theoretical calculations and experimental measurements. We have used Kelvin Probe Force Microscope (KPFM) to measure the energy differences of WSe₂ and Ta₂NiSe₅. The offset of Fermi levels between the WSe₂ and Ta₂NiSe₅ is shown in below **Figure R8**. We have also referred to the energy band arrangement in previous literatures with the same crystal structure and similar thickness. The energy band of WSe₂ has been extensively reported in the literatures and the relative positions of the conduction band minimum (CBM) of multilayer WSe₂ can be directly obtained from previous reports (Nat. Commun. 2021, 12, 7034; ACS Nano 2014, 8, 6265). The CBM of our 60-nm-thick WSe₂ are ~ 4.0 eV, with a bandgap of 1.2 eV. According to experimental studies on the energy band of Ta₂NiSe₅, the CBM of 13-nm thick Ta₂NiSe₅ are ~ 4.6 eV with a bandgap of 0.33 eV (Adv. Funct. Mater. 2022, 32, 2110706). It is also worth noting that the calculated energy band arrangement of the WSe₂/Ta₂NiSe₅ heterojunction by DFT (**Supplementary Fig. 9** of Supplementary Information, as also shown in below **Figure R9**) is **in good agreement with the experimental verifications**, which implies the rationality of the analysis of the energy band diagrams of the WSe₂/Ta₂NiSe₅ heterostructure.

Figure R8 **a**, Surface potential difference between the WSe₂ and Ta₂NiSe₅. **b**, Band diagram of the heterostructure device before contact.

Figure R9 The projected band structures of few-layer WSe₂/Ta₂NiSe₅ heterostructures. **a-f** represents the heterostructures from monolayer 1L Ta₂NiSe₅/1L WSe₂ to six-layer 6L Ta₂NiSe₅/6L WSe₂. The green and red colors indicate the contributions of Ta₂NiSe₅ and WSe₂, respectively. The fermi level is set to 0 eV.

[Actions]:

On Page 10-11 of the Supplementary Information, we have shown the experimental results of the surface potential difference between the WSe₂ and Ta₂NiSe₅ and the band diagram of the heterostructure in **Supplementary Fig. 8** and **Supplementary Fig. 9**.

[Comments 3]: *In the manuscript, the authors said that “the Fermi level of WSe₂ will*

further downshift and the hole concentration is increased, enabling the direct tunneling of the holes from the drain towards the source terminal”. Please provide more explanations. How does the direct tunneling appear between WSe₂ and Ta₂NiSe₅?

[Reply] Thanks for pointing this issue. Regarding this sentence “the Fermi level of WSe₂ will further downshift and the hole concentration is increased, enabling the direct tunneling of the holes from the drain towards the source terminal” in our previous submission, we have found that it may lead to misunderstanding due to the way we described it. To make it clearer, we have revised this sentence to “the Fermi level of WSe₂ would shift downwards, so the hole concentration increases, fostering the tunneling of hole carriers from drain electrode to WSe₂ which are then transported to source electrode.”.

[Actions]:

On Pages 8 and 9 of the revised main text, we have rephrased this sentence to “**the Fermi level of WSe₂ would shift downwards, so the hole concentration increases, fostering the tunneling of hole carriers from drain electrode to WSe₂ which are then transported to source electrode.**”

REVIEWERS' COMMENTS

Reviewer #2 (Remarks to the Author):

The authors have meticulously responded my comments and carefully revised the main text and supporting information. I do not have further comments accordingly. I suggest the editor to consider its publication.

Reviewer #3 (Remarks to the Author):

Question: First, in the aspect of mechanism, most of the current reports are based on either carrier tunneling effect or photogating effect. However, the carrier tunneling effect usually leads to the improvement in response speed while the responsivity changes very little (Nano Lett. 2017, 17, 453; Adv. Mater. 2020, 32, 1902039; Adv. Mater. 2021, 33, 2101449; Adv. Mater. 2021, 33, 2101243). While the photogating effect leads to improved responsivity while the response speed of the device may become several orders of magnitude slower (Nat. Nanotechnol. 2013, 8, 497; Nat. Nanotechnol. 2012, 7, 363; Adv. Funct. Mater. 2016, 26, 1938; Adv. Funct. Mater. 2017, 27, 1603605). The mechanism in article 1 as mentioned by the reviewer (npj 2D Mater. Appl. 2017, 1, 19) belongs to the carrier tunneling effect. It is mentioned that the tunnel layer only plays the role of reducing the density of interface defect states, which achieves an increase in response speed but not significantly improving the responsivity (within one order of magnitude). While the mechanism in article 3 (ACS Appl. Mater. Interfaces 2018, 10, 36512) belongs to the photogating effect. The photogating effect enables improved device responsivity (1 or 2 orders of magnitude), but the response speed is also inevitably reduced by several orders of magnitude. It should be pointed out that the mechanism reported in this manuscript, namely photogating-assisted tunneling, is different from all the previous ones. This novel effect leverages the synergistic interplay between photocarrier multiplication and carrier acceleration through the tunnelling process. As demonstrated in our work, this effect allows simultaneous improvement both in responsivity (by about four orders) and response speed (by about one order), which largely overcome the traditional trade-off between responsivity and response speed. Second, in the aspect of device functionality, the photogating effect in our device shows a dependence on the polarization states of incident light, which can be further tuned by source-drain voltages, allowing for wavelength discrimination with just a two-electrode planar structure. Therefore, the WSe₂/Ta₂NiSe₅ device in our work are no longer limited to the detection of light intensity, but have taken an important step towards the field of multi-functions including intensity, polarization and spectrum. Such a versatile function has been rarely reported in two-port devices as what we report here. Third, it should be also noted that for article 2 mentioned by the reviewer (Adv. Electron. Mater. 2021, 7, 2000920), the reported performance parameters are based on theoretical prediction. The experimental verification of theoretical prediction remains largely unexplored. Based on all above, our work shows significant novelty in terms of new mechanisms and multiple-functions. Importantly, these characteristics help to achieve the long-sought next-generation photodetectors with high responsivity, fast speed, polarization detection, and multi-color

sensing.

Response: I don't think the authors answered my questions. The authors reported photodetectors based on the photogating-assisted tunneling. However, the performance is not better than that based on the photogating effect (For graphene/MoS₂/Si, the responsivity and response time are 3×10⁰⁰⁰ A/W and 500 ns). If the response time of the WSe₂/Ta₂NiSe₅ device reaches up to the nanosecond magnitude, I think this work is enough. Otherwise, this manuscript does not meet the requirements of Nature Communications